# Dietary Sodium Intake Is Positively Associated with Sugar-Sweetened Beverage Consumption in Chinese Children and Adolescents

**DOI:** 10.3390/nu13113949

**Published:** 2021-11-05

**Authors:** Zhenni Zhu, Xueying Cui, Xiaohui Wei, Jiajie Zang, Jingyuan Feng, Zhengyuan Wang, Zehuan Shi

**Affiliations:** 1Division of Health Risk Factors Monitoring and Control, Shanghai Municipal Center for Disease Control and Prevention, 1380 West Zhongshan Road, Shanghai 200036, China; cuixueying@scdc.sh.cn (X.C.); zangjiajie@scdc.sh.cn (J.Z.); wangzhengyuan@scdc.sh.cn (Z.W.); shizehuan@scdc.sh.cn (Z.S.); 2School of Public Health, Fudan University, 130 Dongan Road, Shanghai 200030, China; xhwei16@fudan.edu.cn (X.W.); jyfeng16@fudan.edu.cn (J.F.)

**Keywords:** sugar-sweetened beverage, dietary sodium, salt, total fluid, association

## Abstract

Sugar-sweetened beverage (SSB) consumption among children and adolescents is steadily increasing in China, while the main taste of Chinese food is salty. The present study aimed to determine the relationships between SSB and total fluid consumption and dietary sodium and salt intake among children and adolescents in China. The data were obtained from a cross-sectional investigation in 2015. A total of 3958 participants were included. A 24-h dietary record for three consecutive days was collected to determine the SSB intake and food consumption across school days and rest days. After adjusting for age, sex, yearly household income, maternal education, intentional physical exercise, and instances of eating out in the last week, the dietary sodium intake was positively associated with the SSB consumption (*p* < 0.05), but salt was not. After stratifying by sex, grades, and puberty status, the associations between dietary sodium intake and SSB consumption were significant in girls, in grades 1–5 and before puberty (*p* < 0.05). Dietary sodium intake was positively associated with SSB consumption in Chinese children and adolescents, particularly in young children. A reduction of the sodium intake might help reduce SSB consumption among children and adolescents.

## 1. Introduction

As the dietary pattern of Chinese people has been dramatically Westernized in the past two decades, SSB consumption among children and adolescents has steadily increased [1]. The consumption of sugar-sweetened beverages (SSBs) among children and adolescents is becoming a public health concern due to its adverse effects [2,3]. Excessive SSB consumption has been shown to closely correlate with a higher risk of obesity [4], type-2 diabetes [5], cardiometabolic health [6], and dental caries [7] in children and adolescents. These then contribute to an increased risk of consequent health problems in adulthood [8].

The main taste of Chinese food is salty; accordingly, Chinese people consume a considerable amount of sodium, amounting to more than twice the recommended consumption set by WHO [9,10]. The main sources of dietary sodium in Chinese food are salt, soy sauce, processed food, and monosodium glutamate [11,12]. Overconsumption of dietary sodium is a major risk factor related to elevated blood pressure in children and adolescents, which may, in turn, accelerate the development of hypertension over their whole life course [13].

Excessive dietary sodium intake prompts thirst, which might drive increased consumption of SSBs [14,15]. Indeed, it has been reported that high dietary sodium intake is associated with high consumption of SSBs and increased total fluids needed in children [16,17]. Both SSBs and dietary sodium are risk factors to children’s health, and their negative effects could be compounded by an association between the two. Since dietary sodium intake is generally excessive and SSB consumption has markedly increased recently in China [18], the two could pose double threats to the health of Chinese children. However, little is known about the correlation between SSB consumption and dietary sodium or salt intake among Chinese children and adolescents. Therefore, we performed the present study to determine the relationship between SSB and total fluid consumption and dietary sodium or salt intake among children and adolescents in China.

## 2. Materials and Methods

### 2.1. Study Population

The data in this study were obtained from a cross-sectional investigation that was conducted from September to October 2015 as a part of the Shanghai Diet and Health Survey (SDHS), which was conducted in Shanghai, one of the most developed cities in China. The SDHS was designed to examine the associations of food consumption, energy and nutrient intake, and behavioral factors with nutrition-related health outcomes among local residents. In 2015, a random representative sample of local children aged 6–17 years old who attended school were selected through a multistage stratified sampling method from 20 primary, 20 junior high and 20 senior high (or equivalent secondary vocational) schools (*n* = 4320). The participants who failed to complete either part of the questionnaire survey (*n* = 237), missed anthropometric measurements (*n* = 78), or refused to provide a blood sample (*n* = 47) were excluded. The data from 3958 participants were included in the present analysis. The 2015 SDHS was approved by the Shanghai Municipal Center for Disease Control and Prevention’s Institutional Review Board on 2 September 2015 (no. 2015-15). Written informed consent was obtained from each participant or the participant’s parents or guardians before the survey. The study complied with the code of ethics of the World Medical Association (Declaration of Helsinki) [19].

### 2.2. Dietary Intake

SSBs were defined as nonalcoholic beverages sweetened by sugar, excluding fresh juice. Since the serving sizes of beverages vary in China—for example, a 1.5-L bottle of a drink is a common size and is usually shared by several persons or consumed over several separate occasions—a 24-h dietary record for three consecutive days (including two weekdays and one weekend day) was collected to determine the SSB intake and food consumption across school days and rest days. Each participant was orally instructed to record their daily food intake (including fluid) both at school and home at the beginning and then interviewed face-to-face by interviewers in the consecutive survey days at home. At each survey day, the interviewers collected and checked though the paper, and afterward, revised the food weight and transcribed the draft dietary information into a structured form. Household condiments mainly containing fat or sodium, including cooking oil, salt, soy sauce, and chili sauce, were weighed before and after the three survey days in the same containers. The intake of dietary sodium was estimated according to daily food and condiments consumption using the Chinese food composition database [20,21]. The total fluid intake included all sources of fluid consumed either as a beverage or water in the meal. An SSB non-consumer was defined as someone who reported no SSB intake during the three-day survey period (two weekdays and one weekend day).

Furthermore, the participants were instructed not to change their typical diet or physical activity during the survey period. Their diet records were reviewed by nutrition specialists from local centers of disease control and prevention. The parents or caregivers were interviewed together with the participants who were under 12 years old. No disastrous events, such as heavy rain or snow, affected the normal food supply during the survey period. The weather from approximately September to October 2015 in Shanghai was the typical marine climate of autumn, and the temperature fluctuated within 18–30 °C or 64–86 °F.

### 2.3. Potential Confounders

An individual’s daily food and condiment consumption was calculated from the diet record and household condiment weighing. Information on each participant’s age, school grade, sex, pubertal stage, household income, intentional physical exercise, instances of eating out last week, and maternal education were recorded using an interviewer-administered questionnaire at each participant’s home. The pubertal stage was self-reported by the participants based on menstruation or spermatorrhea. The maternal education level of the participants was reported as years of education. Intentional physical exercise was defined as the physical exercise performed for the purpose of health maintenance or fitness. The household income was the family’s total yearly income (CNY).

### 2.4. Statistical Analyses

Statistical analyses were conducted using the SAS statistical software (v. 9.2; SAS Institute, Cary, NC, USA). The Chi-squared test was applied to determine the differences in the participants’ characteristics. Multivariate linear models were applied to determine the partial correlation coefficients (β) and 95% confidence intervals (CIs) of the differences in participants’ SSB or total fluid consumption versus their dietary salt or sodium intake. Potential confounders—including age, sex, pubertal stage, household income, intentional physical exercise, instances of eating out last week, and maternal education—were introduced as covariates in three different adjusted models. A two-sided *p* < 0.05 was considered to indicate statistical significance.

## 3. Results

### 3.1. Characteristics of the Participants

The final analysis included 3955 participants, consisting of 49.7% boys and 50.3% girls. Of those, 1373 participants consumed SSBs. The proportions of the participants in grades 1–5, 6–9, and 10–12 were 41.3%, 34.5%, and 24.2%, respectively. The percentages of SSB non-consumers and consumers in grades 1–5 were 44.6% and 35.1%, respectively (Table 1).

### 3.2. The Dietary Sources of Sodium and the Correlation between Dietary Salt and Sodium

The leading dietary source of sodium, salt, accounted for 57.4% of the total sodium intake. The following significant sources were soy sauce (13.2%), fungi and algae (6.5%), and monosodium glutamate (4.6%; Figure 1).

After adjusting for age and sex, a significant partial correlation was determined between dietary salt and sodium (0.881, *p* < 0.05; Table 2).

### 3.3. The Association between Dietary Sodium/Salt Intake and SSB Consumption

After adjusting for age, sex, yearly household income, maternal education, intentional physical exercise, and instances of eating out in the last week (Model 3), dietary sodium was positively associated with SSB consumption (*p* < 0.05). With each additional 390 mg of dietary sodium per day, the SSB consumption increased by 1.7 g (95% CI, 0.30–3.10) per day. After stratifying by sex, school grade, and puberty status, the association between dietary sodium intake and SSB consumption was still significant in girls, those in grades 1–5, and those before puberty (*p* < 0.05). With each additional 390 mg of salt intake per day, the SSB consumption increased by 2.7 g (95% CI, 0.99–4.4), 3.5 g (95% CI, 1.49–5.49), and 1.8 g (95% CI, 0.03–3.60) per day, respectively.

After adjusting for age, sex, yearly household income, maternal education, intentional physical exercise, and instances of eating out in the last week (Model 3), there was no association between salt intake and SSB consumption. When looking at specific groups within the overall study population, however, we observed a significant association between salt intake and SSB consumption among participants in grades 1–5 and before puberty (*p* < 0.05). For children in grades 1–5, with each additional 1 g of salt intake per day, the SSB consumption increased by 5.4 g (95% CI, 2.71–8.17) per day. For the children before puberty, with each additional 1 g of salt intake per day, the SSB consumption increased by 3.0 g (95% CI, 0.62–5.35) per day. For the different sexes, there were no significant associations between salt intake and SSB consumption (Table 3).

### 3.4. The Association between Dietary Sodium/Salt Intake and Total Fluid Consumption

After adjusting for the age, sex, yearly household income, maternal education, intentional physical exercise, and instances of eating out in the last week (Model 3), no significant association of dietary sodium intake was found with total fluid consumption in the overall sample. After stratifying by grades, however, significant associations were shown for participants of grades 10–12 (*p* < 0.05); with each additional 390 mg of dietary total sodium per day, their total fluid consumption decreased by 4.8 g (95% CI, −8.98 to −0.59) per day.

After adjusting for the age, sex, yearly household income, maternal education, intentional physical exercise, and instances of eating out in the last week (Model 3), the salt intake was negatively associated with the total fluid consumption (*p* < 0.05). With each additional 1 g of salt intake per day, the total fluid consumption decreased by 4.1 g (95% CI, −7.36 to −0.78) per day. After stratifying by sex, school grade, and puberty status, the same associations between the salt intake and total fluid consumption were still significant in boys, participants in grades 10–12, and those who had entered puberty (*p* < 0.05). With each additional 1 g of salt intake per day, the total fluid consumption decreased by 7.8 g (95% CI, −12.72 to −2.81), 9.6 g (95% CI, −16.31 to −2.81), and 7.7 g (95% CI, −13.23 to −2.18) per day, respectively (Table 4).

## 4. Discussion

In this study, we found that dietary sodium intake was positively associated with SSB consumption among the study children and adolescents, but dietary salt intake was not. Though dietary sodium and salt were significantly correlated with each other in our results, the two exhibited different associations with SSB consumption. For instance, though it is generally known that Chinese people consume a considerable amount of dietary salt [10], and though previous studies showed that dietary salt was associated with the consumption of SSBs in Western countries [17,22], salt was found to contribute less than two-thirds of dietary sodium in the current participants, which is in line with the finding of a previous study that showed that salt contributed approximately two-thirds of Chinese people’s sodium intake [11]. So, the current participants were recruited in Shanghai, a southern metropolis in China where salty sauces are commonly used in food preparation, but their salt intake might not give a complete picture of their overall sodium intake, e.g., due to salty seasonings, such as soy sauce, applied to their food. This might explain why no association was found between dietary salt and SSB consumption in the current study. As such, this study indicates that health promotion in China should not only focus on the salt intake but also on the total sodium intake, especially with regard to salty seasonings, which contribute a considerable part of people’s dietary sodium.

Compared with studies on Western children, the absolute additional SSB consumption for each 390 mg/d of dietary sodium intake was not remarkable in the current study. This could be attributed to Chinese children’s absolute SSB consumption remaining lower, at present, than that of their Western counterparts [6,23]. Yet, the finding is far from a relief, as based on the trends of Western countries, the SSB consumption is expected to continue to steadily rise among Chinese children, and the adverse effects of that should never be underestimated. Moreover, children are in critical period of taste formation that could last for their whole life [24], and a tendency to enjoy heavy tastes of salty and sweet flavors is not favorable for health maintenance.

When the participants were stratified by school grades, we discovered that the dietary sodium intake, as well as the salt intake, was significantly positively associated with SSB consumption among children in grades 1–5 (6–10 y) but not those in the senior grades. Our findings coincide with the understanding that the linkage between dietary sodium and SSB consumption is significant at a young age but not in adulthood [25]. As such, young children are more vulnerable to the adverse consequences of SSB overconsumption [26]. With that in mind, we propose that they should be the first priority of health promotions targeting SSB or salt reduction among the general population.

When stratified by sex, the dietary sodium intake was associated with SSB consumption among the girls but not the boys in our study, though it was reported that sweet and salty taste preferences were positively correlated among all the children [27]. Since boys usually consumed more SSBs than the girls did [6], we speculate that the desire for SSBs among boys may stem, at least in part, from non-dietary factors such as sense of identity, while girls may consume SSBs more due to a taste desire after salty food intake. This might explain the sex difference found in the association between SSB consumption and sodium intake.

Further to this, we found that dietary sodium and salt were differently associated with the total fluid consumption. The dietary sodium intake was not associated with the total fluid consumption, but the salt intake was negatively associated with total fluid consumption in our participants. This was the opposite finding to previous studies that suggested dietary sodium was positively correlated with the total fluid consumption [17]. Yet, the absolute difference was not that notable in the current results, i.e., a difference of 1 g salt intake was only related to a difference of 4 mL in total fluid consumption. We suggest that, as mentioned above, the salt intake might not fully represent the sodium intake since more than one-thirds of the dietary sodium originated from sources other than salt in the current study population. For that reason, we are inclined to take the relationship of dietary sodium with total fluid consumption into account (i.e., no association), rather than the association between the dietary salt intake and total fluid consumption.

One limitation of this study was the methodology used to assess the food and beverage consumption. A dietary record was collected to determine the food and beverage consumption of each participant. In this way, the food name and consumed amount were self-reported by the participants; thus, the food consumption was limited by the accuracy of participants’ estimation and recall. Furthermore, although we adjusted for several potential confounding factors, we did not include other unknown confounders that might have caused bias in the current results. Finally, it is logical to hypothesize that excessive sodium intake triggers the desire for sweet beverages, but the cross-sectional nature of the current study did not allow us to infer causal associations.

## 5. Conclusions

The dietary sodium intake was associated with SSB consumption in Chinese children and adolescents, particularly in young children. The total dietary sodium intake was found to be a more objective factor to focus on than the salt intake only in health promotions. We propose that since the relationship between the dietary sodium intake and SSB consumption, a reduction of the sodium intake might reduce SSB consumption among children and adolescents.

## Figures and Tables

**Figure 1 nutrients-13-03949-f001:**
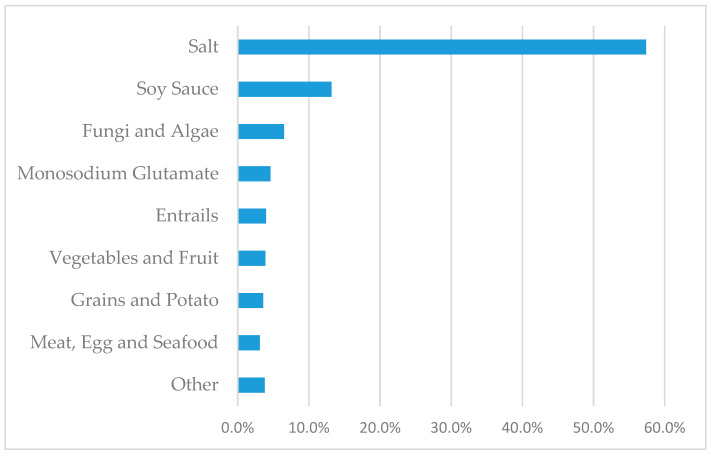
Dietary sources of sodium among 3958 participants from SDHS in 2015.

**Table 1 nutrients-13-03949-t001:** Characteristics of the participants.

		SSB Intake		*p*
	All	Non-Consumers ^a^	Consumers	
N (%)	3955 (100.0)	2582 (65.3)	1373 (34.7)	
Sex, %				0.375
Boys	49.7	49.2	50.6	
Girls	50.3	50.9	49.4	
Grade, %				<0.001
1–5 (6–10 y)	41.3	44.6	35.1	
6–9 (11–14 y)	34.5	35.3	33.1	
10–12 (15–17 y)	24.2	20.1	31.8	
Entered puberty, %				<0.001
Entered puberty	37.0	33.6	43.4	
Not entered puberty	63.0	66.4	56.6	
Yearly household income, %				0.201
Above average (>60,000 CNY)	33.2	33	33.6	
Average (30,000–59,999 CNY)	23.9	24.1	23.4	
Below average (<30,000 CNY)	21.8	22.6	20.3	
No answer	21.1	20.3	22.7	
Intentional physical exercise, %				0.819
No	46.6	46.7	46.3	
Yes	53.5	53.3	53.7	
Instances of eating out last week, %				<0.001
0	57	60.6	50.3	
1–2	25.1	23.7	27.8	
≥3	17.9	15.8	21.9	
Maternal education, year (SD)	11.8 (4.0)	11.8 (4.0)	12.0 (4.0)	0.115
Dietary salt, g/d (SD)	6.4 (4.1)	6.2 (3.9)	6.7 (4.4)	<0.001
Dietary sodium, mg/d (SD)	4297.6 (2285.5)	4160.9 (2267.2)	4554.5 (2298.4)	<0.001
Dietary sodium equivalent to salt, g/d (SD)	11 (5.9)	10.7 (5.8)	11.7 (5.9)	<0.001
SSB consumption, g/d (SD)	59.4 (126.3)	0.0 (0.0)	171.1 (163.8)	<0.001
Total fluid consumption, g/d (SD)	750 (428.3)	708.5 (403.9)	827.7 (460.8)	<0.001

^a^ In this study, a non-consumer was defined as someone who reported no SSB intake during the three-day survey period (two weekdays and one weekend day).

**Table 2 nutrients-13-03949-t002:** Partial correlation coefficient between dietary sodium and salt.

	Dietary Sodium	Dietary Salt
Dietary sodium	1	0.881 ^a^
Dietary salt		1

^a^ represents *p* < 0.05. Adjusted by age and sex.

**Table 3 nutrients-13-03949-t003:** Multiple linear regression analysis of SSB consumption (g/d) and dietary sodium intake (390 mg/d)/salt intake (1 g/d) among the SSB consumers ^a,b^.

		Sodium			Salt		
		β	95% CI	*p*	β	95% CI	*p*
SSB consumers (*n* = 1373)							
Model 1 ^c^		2.13	(0.67, 3.59)	0.004	1.25	(−0.72, 3.24)	0.213
Model 2 ^c^		1.69	(0.29, 3.09)	0.018	0.93	(−0.97, 2.84)	0.336
Model 3 ^c^		1.70	(0.30, 3.10)	0.017	0.86	(−1.04, 2.77)	0.373
Stratified by grades							
Model 1	Grade 1–5 (6–10 y)	3.67	(1.74, 5.57)	<0.001	5.53	(2.96, 8.09)	<0.001
	Grade 6–9 (11–14 y)	1.32	(−1.11, 3.74)	0.287	−0.20	(−3.32, 2.93)	0.902
	Grade 10–12 (15–17 y)	2.29	(−0.83, 5.42)	0.149	0.42	(−4.06, 4.91)	0.853
Model 2	Grade 1–5 (6–10 y)	3.49	(1.49, 5.48)	<0.001	5.44	(2.71, 8.16)	<0.001
	Grade 6–9 (11–14 y)	1.15	(−1.15, 3.45)	0.325	−0.17	(−3.13, 2.80)	0.913
	Grade 10–12 (15–17 y)	1.27	(−1.76, 4.30)	0.410	−0.87	(−5.23, 3.49)	0.696
Model 3	Grade 1–5 (6–10 y)	3.49	(1.49, 5.49)	<0.001	5.44	(2.71, 8.17)	<0.001
	Grade 6–9 (11–14 y)	1.00	(−1.29, 3.30)	0.390	−0.42	(−3.38, 2.54)	0.781
	Grade 10–12 (15–17 y)	1.17	(−1.88, 4.22)	0.451	−1.27	(−5.63, 3.09)	0.567
Stratified by puberty status							
Model 1	Entered puberty	1.43	(−0.86, 3.73)	0.220	−1.09	(−4.24, 2.06)	0.498
	Not entered puberty	2.51	(0.57, 4.45)	0.011	3.40	(0.86, 5.94)	0.009
Model 2	Entered puberty	1.51	(−0.76, 3.78)	0.193	−0.77	(−3.89, 2.35)	0.629
	Not entered puberty	1.78	(−0.01, 3.57)	0.051	2.98	(0.61, 5.34)	0.014
Model 3	Entered puberty	1.50	(−0.78, 3.78)	0.196	−0.90	(−4.04, 2.23)	0.572
	Not entered puberty	1.81	(0.03, 3.60)	0.047	2.98	(0.62, 5.35)	0.014
Stratified by sex							
Model 1	Boys	1.24	(−1.08, 3.57)	0.295	0.13	(−2.91, 3.16)	0.935
	Girls	2.80	(1.05, 4.55)	0.002	2.33	(−0.14, 4.81)	0.065
Model 2	Boys	0.52	(−1.69, 2.73)	0.644	−0.43	(−3.33, 2.46)	0.768
	Girls	2.65	(0.94, 4.36)	0.002	2.33	(−0.10, 4.76)	0.060
Model 3	Boys	0.57	(−1.64, 2.79)	0.610	−0.52	(−3.41, 2.37)	0.726
	Girls	2.69	(0.99, 4.40)	0.002	2.31	(−0.12, 4.74)	0.062

^a^ In all models, the SSB consumption was treated as the independent variable and the sodium or salt intake was treated as the dependent variable. ^b^ β represents the partial correlation coefficients in the models, which means the amount of SSB consumed for each additional 390 mg sodium or 1 g salt intake. ^c^ Model 1: Adjusted by age and sex; Model 2: Model 1 further adjusted by yearly household income (categorical variable) and maternal education (continuous variable); Model 3: Model 2 further adjusted by intentional physical exercise (categorical variable) and number of instances of eating out last week (categorical variable).

**Table 4 nutrients-13-03949-t004:** Multiple linear regression analysis of total fluid consumption (g/d) and dietary sodium intake (390 mg/d)/salt intake (1 g/d) among all participants ^a,b^.

		Sodium			Salt		
		β	95% CI	*p*	β	95% CI	*p*
All (*n* = 3955)							
Model 1 ^c^		−0.20	(−2.45, 2.05)	0.859	−4.92	(−8.17, −1.67)	0.003
Model 2 ^c^		−0.51	(−2.79, 1.76)	0.659	−4.01	(−7.30, −0.72)	0.017
Model 3 ^c^		−0.47	(−2.74, 1.81)	0.688	−4.07	(−7.36, −0.78)	0.015
Stratified by grades							
Model 1	Grade 1–5 (6–10 y)	2.49	(−0.98, 5.95)	0.160	−2.11	(−6.81, 2.59)	0.379
	Grade 6–9 (11–14 y)	1.27	(−3.02, 5.57)	0.560	−1.72	(−7.49, 4.04)	0.557
	Grade 10–12 (15–17 y)	−3.56	(−7.71, 0.58)	0.092	−8.93	(−15.66, −2.21)	0.009
Model 2	Grade 1–5 (6–10 y)	3.65	(0.07, 7.24)	0.046	−0.33	(−5.27, 4.61)	0.895
	Grade 6–9 (11–14 y)	1.62	(−2.75, 5.99)	0.467	−0.80	(−6.67, 5.08)	0.791
	Grade 10–12 (15–17 y)	−4.93	(−9.12, −0.74)	0.021	−9.52	(−16.27, −2.76)	0.006
Model 3	Grade 1–5 (6–10 y)	3.53	(−0.06, 7.12)	0.054	−0.49	(−5.43, 4.44)	0.844
	Grade 6–9 (11–14 y)	1.47	(−2.91, 5.86)	0.510	−0.98	(−6.86, 4.91)	0.745
	Grade 10–12 (15–17 y)	−4.78	(−8.98, −0.59)	0.026	−9.56	(−16.31, −2.81)	0.006
Stratified by puberty status							
Model 1	Entered puberty	−2.10	(−5.71, 1.51)	0.254	−8.16	(−13.66, −2.67)	0.004
	Not entered puberty	0.72	(−2.22, 3.66)	0.631	−3.15	(−7.18, 0.87)	0.124
Model 2	Entered puberty	−3.18	(−6.83, 0.48)	0.089	−7.57	(−13.10, −2.04)	0.007
	Not entered puberty	1.15	(−1.82, 4.13)	0.447	−1.79	(−5.90, 2.32)	0.393
Model 3	Entered puberty	−3.02	(−6.68, 0.64)	0.105	−7.70	(−13.23, −2.18)	0.006
	Not entered puberty	1.10	(−1.87, 4.08)	0.468	−1.85	(−5.96, 2.26)	0.377
Stratified by sex							
Model 1	Boys	−2.09	(−5.56, 1.39)	0.239	−8.22	(−13.12, −3.31)	0.001
	Girls	1.09	(−1.77, 3.95)	0.454	−2.21	(−6.44, 2.02)	0.305
Model 2	Boys	−2.99	(−6.49, 0.51)	0.094	−7.72	(−12.68, −2.76)	0.002
	Girls	1.25	(−1.67, 4.18)	0.400	−0.91	(−5.20, 3.39)	0.679
Model 3	Boys	−2.87	(−6.37, 0.62)	0.107	−7.77	(−12.72, −2.81)	0.002
	Girls	1.29	(−1.63, 4.21)	0.388	−0.93	(−5.22, 3.37)	0.672

^a^ In all models, total fluid consumption was treated as the independent variable and the sodium or salt intake was treated as the dependent variable. ^b^ β represents the partial correlation coefficients in the model, which means the amount of total fluid consumed for each additional 390 mg sodium or 1 g salt intake. ^c^ Model 1: Adjusted by age and sex; Model 2: Model 1 further adjusted by yearly household income (categorical variable) and maternal education (continuous variable); Model 3: Model 2 further adjusted by intentional physical exercise (categorical variable) and number of instances of eating out last week (categorical variable).

## Data Availability

The datasets used and analyzed in the current study are available from the corresponding author on reasonable request.

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
