# Peer review of "Dietary Sodium Intake Is Positively Associated with Sugar-Sweetened Beverage Consumption in Chinese Children and Adolescents"

_nutrients, 2021, doi:10.3390/nu13113949_

Round 1
Reviewer 1 Report
The article 'Dietary sodium intake is positively associated with sugarsweetened beverage consumption in Chinese children and adolescents' is an original contribution to science, taking a novel approach to sodium intake among young people. The selection of the study group takes into account the economic and social context in an accurate and relevant way.
The authors correctly suggest paying attention to sodium intake in the diet due to its extensive use in Chinese cuisine.
- Abstract
- „household yearly income” change to „yearly household income” in the abstract and elsewhere in the work
- Sentence “Dietary sodium intake was positively associated with SSB consumption in Chinese children and adolescents, particularly in young children”. “in” may be unnecessary in this sentence. Consider removing it.
- Introduction:
- “Differing from Western food, the main sources of dietary sodium in Chinese food are salt, soy sauce, processed food, and monosodium glutamate [11, 12].”
The subordinate phrase Differing from Western food does not appear to be modifying the subject the main sources of dietary sodium in Chinese food. Rewrite the sentence to avoid a dangling modifier.
- Materials and Methods
- “In 2015, a random representative sample of local children aged 6–17 years old who attended school was selected through a multistage stratified sampling method from 20 primary, 20 junior high and 20 senior high (or equivalent secondary vocational) schools (n = 4320)”
were selected
2. “Written informed consent was obtained from each participant or the participant's parents or guardian before the survey”
guardians
3. Think to add the citation to the World Medical Association Declaration of Helsinki
- Results
- incorrectly rounded results in table 1, for boys and girls.
- “The next greatest sources were soy sauce (13.2%), fungi and algae (6.5%), and monosodium glutamate”
Think to change, e.g. The following significant sources were soy sauce…
3. Figure 1 “Meat,Egg and Seafood” Consider adding a space.
4. I propose to change the format of Table 3 due to the difficulty in reading and understanding its content
5. “With each additional 1g of salt intake per day, the total fluid consumption decreased by 7.8g (95% CI, -12.72 to -2.81), 9.6g (95% CI, -16.31 to -2.81), and 7.7g (95% CI, -13.23 to -2.18) per day, respectively (.
).” brackets left at the end of the sentence
- Discussion
- “corelated" change to correlated
- “Yet, the absolute difference was not that notable in the current results, i.e., a difference of 1 g salt intake was only related with a difference of 4 ml in total fluid consumption.” …to a difference of 4 ml in total fluid consumption
- Conclusion
- write sentence about beneficial among for children and adolescents after reduction of the sodium in conclusion.
- References
- Please correct the bibliographies as required on the website.
You used two different styles of apostrophes or quotation marks in your document. Both styles are acceptable, but it’s best to be consistent.
Try to avoid frequent repetitions in the text by using synonyms.
Complete the articles: a, an and the in the text
Reviewer 2 Report
Introduction:
- You use sodium and salt intake interchangeably in the introduction: “Therefore, we performed the present study to determine the relationship between SSB and total fluid consumption and dietary sodium or salt intake among children and adolescents in China” however, you clearly looked at both. Why is it important to look at total dietary sodium and salt intake separately?
Materials and Methods:
- Is there a reason why you mentioned anthropometric measurements, but do not present any data for them or include them in the analysis?
- How was physical activity assessed?
- How was the 24-hour dietary record collected? You mentioned self-reported in the discussion, but not in the methods.
- How was an individual’s daily condiment consumption calculated from the 3-day household condiment consumption?
- How did you differentiate between dietary sodium and salt intake in the food recalls? Overall, the definitions and methods used to discern between dietary sodium and salt are unclear. This makes it difficult to interrupt the results.
Results:
- In all of the models you stratified by grade. It would be nice to see the Table 1 characteristics especially dietary sodium, SSB and fluid intake stratified by grade, just a suggestion.
- In Table 1, what is the difference between dietary sodium and dietary salt and why are they in different units? Also, you didn't explain why you reported the dietary sodium equivalent.
Discussion:
- You mentioned in the discussion that you adjusted for daily energy intake, however, I did not see where. Additionally, how did you adjust for SSB intake in the total energy intake?
